# Numerical Simulation of Hydrodynamics and Sediment Transport in the Surf and Swash Zone Using OpenFOAM®

**Ioannis Kazakis and Theophanis V. Karambas \***

Department of Civil Engineering, Aristotle University of Thessaloniki, University Campus,
54 124 Thessaloniki, Greece
**\*** Correspondence: karambas@civil.auth.gr; Tel.: +30-2310995805

**Abstract:** This study focuses on the numerical investigation of the 3D hydrodynamic processes of coastal zones such as wave breaking, wave-induced currents, and sediment transport, using the multiphase, interFoam solver of OpenFOAM® (a state-of-the-art, open-source CFD numerical tool). The numerical scheme is suitably framed by initial conditions of wave propagation and absorption using waves2Foam wave library. The turbulence closure problem is handled using a buoyancy modified $k$–$\omega$ SST model. In order to predict the sediment transport rate due to waves and currents (bed load, sheet flow, and suspended load over ripples), a transport-rate formula involving unsteady aspects of the sand transport phenomenon is implemented. For the suspended load in the surf zone, the Bailard formula is adopted after considering that the dissipation mechanism is the wave breaking. Results concerning wave height, longshore current, turbulence kinetic energy, and sediment transport are compared against experimental data and semi-empirical expressions.

**Keywords:** hydrodynamics; sediment transport; surf zone; swash zone; OpenFOAM®

## 1. Introduction

Protection and restoration of coasts and sandy beaches are related to the hydrodynamic and morphodynamic phenomena. The wave breaking and the related hydrodynamic processes leads to the presence of strong wave-induced currents and the corresponding morphodynamic processes. Sediment transport results in the accretion /erosion of coastal areas, with direct effects on the morphological formation of the coastal zone along with economic and social impacts.

The use of Reynolds–Averaged Navier Stokes (RANS) equations coupled with the volume of fluid (VOF) method for tracking free-surface position are widely used by several researchers in the numerical modeling of coastal engineering processes [1–3].

The widely used CFD package OpenFOAM® contains a method for solving free-surface Newtonian flows using the Reynolds averaged Navier–Stokes equations coupled with volume of fluid method (VOF) and is becoming very popular across coastal engineering research in recent years as the development of wave libraries waves2Foam [4] and OlaFlow [5–7], both based on the multiphase interFoam solver (part of OpenFOAM®), enhanced the ability of the popular CFD computational tool to predict wave phenomena in coastal engineering problems

Both waves2Foam and OlaFlow have been used mainly in two-dimensional applications to verify and investigate the effectiveness of interFoam solver parameters such as space and time discretization, numerical scheme options, and performance [8], or the implementation and effectiveness of available turbulence models [9] in the hydrodynamics of surf zones. Additionally, interFoam coupled with the above wave generation and absorption libraries is collocated with realistic coastal zone applications [5,6,10,11]. The numerical scheme is further enriched with boundary conditions that take into account the interaction of irregular waves with two-dimensional permeable coastal structures [7,12].

Obtained results generally seem to predict wave characteristics with accuracy. Nevertheless, relative instabilities of wave height as wave train propagates and an overestimation of wave damping rate are observed when comparing against experimental and analytical data [8]. Besides this, overproduction of turbulent kinetic energy takes place even in regions of nearly potential flow as a common phenomenon in RANS model applications of water-wave propagation when using a turbulence model such as $k$–$\omega$ or $k$–$\omega$ SST [13].

Mayer and Madsen [14] have demonstrated the instability exhibited by the $k$–$\omega$ turbulence model in potential flow regions leading to the overproduction of turbulent kinetic energy. With appropriate manipulations they have been able to significantly eliminate the increased value of the turbulent viscosity, indicating the direction for further study of the problem. Brown et al. [9] applied various turbulence models and DNS simulations using interFoam to simulate spilling and plunging breakers indicating each turbulence model features.

Devolder et al. [15] observed the same phenomenon of overproduction of turbulence and assumed that this was due to the high values of turbulent viscosity in the area of the air-water interface. To limit the overproduction of turbulent kinetic energy they introduced an additional term in the equation of turbulence kinetic energy, which fulfills the amount of energy loss caused by TKE's overproduction. According to Larsen and Fuhrman (2018) [13], this is an effective bypass method, but it fails to manage the specific cause of the phenomenon. Instead, they extend the theory of Mayer and Madsen [14] considering mathematical conditions for limiting turbulent viscosity production, both for the potential flow regions and surf zone.

Breaking wave-induced currents in the surf and swash zone can be simulated by using nonlinear wave breaking and non-breaking wave propagation models (such as the above type of models or the Boussinesq ones [6,16]). These models, as part of the simulation of irregular non-linear wave propagation, i.e., refraction, shoaling, diffraction, nonlinear wave-wave interactions, reflection, (presence of structures), breaking, and runup, are able to reproduce automatically in the current field [6,16]. On the other hand, sediment transport prediction in the nearshore region (included swash zone) requires the use of advanced wave modeling to provide information on the wave asymmetry, swash zone hydrodynamics, turbulence kinetic energy and dissipation rate, 3D structure of the current, etc. [16,17].

The common characteristic of the above applications of the interFoam two-phase flow solver for simulating coastal zone wave processes, according to author's knowledge, is that they are limited in two spatial dimensions (one horizontal and one vertical) applications. The present work aims to demonstrate the effectiveness of the open source (CFD) computational package, OpenFOAM®, by assessing the hydrodynamic processes of the coastal zone through the reproduction of benchmark experimental cases concerning wave phenomena of coastal zones such as wave-induced currents [18,19], taking into account their three-dimensional features. The flow-simulation unit is based on the numerical solution of three-dimensional Navier-Stokes equations using the multi-phase interFoam solver, part of OpenFOAM®. The generation and absorption of waves is achieved with the contribution of the waves2Foam library [4,20], while the modified $k$–$\omega$ SST numerical model developed by Devolder et al. [15] is used to simulate turbulent flow characteristics. The results of the work of Larsen et al. [8] are considered for the parameterization of the numerical model. The derived hydrodynamic results concerning wave height, turbulent kinetic energy, and longshore current are used to predict longshore sediment transport rates using the mathematical formula of Dibajnia and Watanabe [21–23].

This paper is structured as follows. After this introductory part, governing equations for the hydrodynamic numerical model, the numerical setup, and the mathematical formula for sediment transport prediction are presented (Sections 2–4). Subsequently, in Section 5, validation cases concerning the nearshore hydrodynamic processes as well sediment transport due to waves and wave-induced currents are presented. The numerical results are compared with experimental measurements. In Section 6 an application in a realistic dimension sloping beach is presented in order to investigate the longshore sedi-

ment transport rates under irregular waves. The numerical results are compared with a well-known empirical formula. Finally, the conclusions are drawn in Section 7.

## 2. Governing Equations

### 2.1. Hydrodynamics

The flow equations for two incompressible, isothermal immiscible fluids in three dimensions consist of the time-dependent continuity equation for conservation of mass and the three time-dependent conservation of momentum equations. The set of equations is given below. In a Cartesian coordinate system, in $x, y, z$ dimension, respectively, the Reynolds averaged Navier-Stokes equations are written:

$$\left( \frac{\partial u}{\partial x} + \frac{\partial v}{\partial y} + \frac{\partial w}{\partial z} \right) = 0 \tag{1}$$

$$\frac{\partial \rho u}{\partial t} + \left( u\frac{\partial \rho u}{\partial x} + v\frac{\partial \rho u}{\partial y} + w\frac{\partial \rho u}{\partial z} \right) = -\frac{\partial p^*}{\partial x} + \mu_{eff}\left( \frac{\partial^2 u}{\partial x^2} + \frac{\partial^2 u}{\partial y^2} + \frac{\partial^2 u}{\partial z^2} \right) - g_z z\frac{\partial \rho}{\partial x} + \sigma\kappa\frac{\partial \alpha}{\partial x} \tag{2}$$

$$\frac{\partial \rho v}{\partial t} + \left( u\frac{\partial \rho v}{\partial x} + v\frac{\partial \rho v}{\partial y} + w\frac{\partial \rho v}{\partial z} \right) = -\frac{\partial p^*}{\partial y} + \mu_{eff}\left( \frac{\partial^2 v}{\partial x^2} + \frac{\partial^2 v}{\partial y^2} + \frac{\partial^2 v}{\partial z^2} \right) - g_z z\frac{\partial \rho}{\partial y} + \sigma\kappa\frac{\partial \alpha}{\partial y} \tag{3}$$

$$\frac{\partial \rho w}{\partial t} + \left( u\frac{\partial \rho w}{\partial x} + v\frac{\partial \rho w}{\partial y} + w\frac{\partial \rho w}{\partial z} \right) = -\frac{\partial p^*}{\partial z} + \mu_{eff}\left( \frac{\partial^2 w}{\partial x^2} + \frac{\partial^2 w}{\partial y^2} + \frac{\partial^2 w}{\partial z^2} \right) + g_z z\frac{\partial \rho}{\partial z} + \sigma\kappa\frac{\partial \alpha}{\partial z} \tag{4}$$

In which ρ is the density of the fluids $\rho_{water}$ and $\rho_{air}$, $\vec{g}_z = \vec{g} = [0, 0, -9.81] \text{ m/s}^2$, $p^*$ is the pseudo-dynamic pressure which is equal to the total pressure excess of the hydrostatic, $p^* = p - \rho g \cdot x$, where $\mathbf{x} = $ x $\rightarrow$ is the Cartesian coordinate vector (x, y, z), $\mu_{\text{eff}}$ is the effective dynamic viscosity, where $\mu_{eff} = \mu + \mu_t = \mu + \rho v_t$, where μ is the dynamic molecular viscosity, $\mu_t$ is the dynamic eddy viscosity, and $v_t$ is the turbulent viscosity given by the chosen turbulence model. Last term of Equations (2)–(4) denotes the effect of surface tension tensor as the two previous terms come for external body forces (including gravity) and mean-strain rate tensor, respectively. Briefly, all symbols with their SI units are given in Table 1.

**Table 1.** Flow equation's quantities and related SI units.

| Symbol | Name | Unit |
|---|---|---|
| $\rho$ | Density | kg·m$^{-3}$ |
| $t$ | Time | [s] |
| $p^*$ | P$_{\text{total}}$ - P$_{\text{hydrostatic}}$ | Pa |
| $g$ | Gravitational acceleration | m·s$^{-2}$ |
| $\mu$ | Dynamic Viscosity | kg·m$^{-1}$·s$^{-1}$ |
| $v_t$ | Turbulent Viscosity | m$^2$·s$^{-1}$ |
| $u, v, w$ | Velocity components at $x, y, z$ Directions | m ·s$^{-1}$ |
| $\sigma$ | Surface tension | Nm$^{-1}$ |
| $\kappa$ | Curvature | m$^{-1}$ |
| $a$ | Volume fraction | $a \in [0, 1]$ |

The surface tension coefficient between air and water at 20 °C is 0.074 kg/s$^2$. As observed in Larsen et al. (2019) [8], switching this parameter between zero and its physical value has no effect on such kind of simulations, as coastal engineering application appertain to gravity wave regime.

The above set of equations is coupled with the volume of fluid method (VOF) for tracking free-surface position, as proposed by Hirt and Nichols [1,24]. When modeling the flow by this method, an important role is played by the determination of the boundary between the phases (water-air or air-water). According to the basic idea of the volume of fluid method, for each computational cell there is a certain scalar quantity ($\alpha$) representing the filling degree of a given cell with one phase; for example, water. If in some cells this

value is 0, then it is empty; if equal to 1, then it is filled; if its value lies between 0 and 1, then we can say that the cell contains a free (interphase) boundary. In other words, the volume fraction of water $\alpha$ is defined as the ratio of the water volume in a cell to the total volume of a given cell. Accordingly, the quantity $1 - \alpha$ represents the volume fraction of the second phase in a given cell-air. At the initial moment t = 0 of time, the distribution of the field of this quantity is given $\alpha = 0$ and its further temporal evolution is calculated as a solution of the transport equation [1]. Because a sharp interface must be maintained and a value must be conserved and bounded between 0 and 1, especially in OpenFOAM®, an artificial compression term is adopted according to [24] that reduces the dissipative nature of the interface when compared to the first implementation of the VOF method of Hirt and Nichols [1].

*2.2. Turbulence Modelling*

A turbulence model is required for the simulation of turbulent characteristics of the flow. OpenFOAM® provides the ability to use and modify benchmark turbulence models such as $k$–$\varepsilon$, $k$–$\omega$ $k$–$\omega$ SST etc. Higuera et al. [7] and Jacobsen et al. [4] used the $k$–$\omega$ SST turbulence model to simulate the turbulent characteristics of the flow.

On the other hand, Larsen and Fuhrman [13] had detected wave energy loss due to the over-production of turbulence, a phenomenon that was observed even in regions with potential flow characteristics and resulted to underestimations in the measured wave height. An extended research concerning the effectiveness of these turbulence numerical models under the influence of spilling and plunging breakers was carried out by Brown et al. [9]. The performance of each turbulent model was ascertained in combination with the implementation of the interFoam solver.

Thus, in this work, turbulence modeling is achieved using the incompressible $k$–$\omega$ SST model as implemented by Menter [25] with the addition of a buoyancy term by Devolder et al. [14]. The buoyancy modified $k$–$\omega$ SST turbulence model differs from the original $k$–$\omega$ SST model as provided in OpenFOAM® because the density is explicitly included in the turbulence transport equations and a buoyancy term is added to the turbulent kinetic energy (TKE) equation.

The two equations of the buoyancy modified $k - \omega$ SST model are defined as:

$$\frac{\partial \rho k}{\partial t} + \frac{\partial \rho k u_i}{\partial x_i} - \frac{\partial}{\partial x_i}\left[\rho(\nu + \sigma_k \nu_t)\frac{\partial k}{\partial x_i}\right] = \rho P_k + G_b - \rho \beta^* \omega k \tag{5}$$

$$\frac{\partial \rho \omega}{\partial t} + \frac{\partial \rho \omega u_i}{\partial x_i} - \frac{\partial}{\partial x_i}\left[\rho(\nu + \sigma_\omega \nu_t)\frac{\partial \omega}{\partial x_i}\right] = \frac{\gamma}{\nu_t}\rho G - \rho \beta \omega^2 + 2(1 - F_1)\rho\frac{\sigma_{\omega 2}}{\omega}\frac{\partial k}{\partial x_i}\frac{\partial \omega}{\partial x_i} \tag{6}$$

$$G = \nu_t \frac{\partial u_i}{\partial x_j}\left(\frac{\partial u_i}{\partial x_j} + \frac{\partial u_j}{\partial x_i}\right) \tag{7}$$

$$\nu_t = \frac{a_1 k}{max(a_1 \omega, SF_2)} \tag{8}$$

where k is the turbulent kinetic energy, $P_k$ is the production term of k ($P_k = \min(G, 10\beta^* k\omega)$), $\nu$ is the kinematic viscosity, $\nu_t$ is the turbulent kinematic viscosity, $\omega$ is the specific dissipation rate, and $S$ is the mean rate of strain of the flow;

where $F_1$ and $F_2$ are blending functions:

$$F_1 = tanh\left\{min\left[max\left(\frac{\sqrt{k}}{\beta^* \omega y}, \frac{500\nu}{y^2 \omega}\right), \frac{4\sigma_{\omega 2} k}{CD_{k\omega} y^2}\right]\right\}^4 \tag{9}$$

$$CD_{k\omega} = max\left(2\rho\frac{\sigma_{\omega 2}}{\omega}\frac{\partial k}{\partial x_i}\frac{\partial \omega}{\partial x_i}, 10^{-10}\right)$$

$$F_2 = tanh\left[\left[max\left(\frac{2\sqrt{k}}{\beta^* \omega y}, \frac{500\nu}{y^2 \omega}\right)\right]^2\right] \tag{10}$$

The buoyancy term $G_b$ is defined as:

$$G_b = -\frac{\nu_t}{\sigma_t}\frac{\partial \rho}{\partial x_i}g_i \tag{11}$$

The following values of the above coefficients are adopted:

$$\alpha_1 = 0.31, \ \gamma_1 = 0.55, \ \sigma_{\kappa 1} = 0.85, \ \sigma_{\omega 1} = 0.5, \ \beta_1 = 0.075$$
$$\beta^* = 0.09, \ \gamma_2 = 0.44, \ \sigma_{\kappa 2} = 1, \ \sigma_{\omega 2} = 0.856, \ \beta_2 = 0.0828$$

The buoyancy term according to Devolder et al. [14] contributes to the suppression of turbulence level at the free-water surface especially in zones where the governing direction of the density gradient is vertical, i.e., the zone near the free surface where non–breaking waves are propagating and consequently the turbulent viscosity $\nu_t$ tends to zero. As a result, in case of non-breaking waves the model switches to a laminar regime near the free surface, preventing excessive wave damping. On the contrary, a fully turbulent solution is obtained in the surf zone where the density gradient consists of an important horizontal component. At the breaking point, this condition is obtained when shoaling waves are reaching their limiting wave height.

### 3. Numerical Setup

#### 3.1. Boundary and Initial Condition

The set of equations is framed by the appropriate boundary and initial conditions for each field variable, water-air fraction *alpha.water*, pressure *p_rgh*, velocity *U*, turbulence kinetic Energy TKE *k*, turbulence viscosity nut and dissipation rate of TKE *omega*. At the inlet and outlet boundaries of the numerical domain, wave generation and absorption conditions were used respectively, as relaxation zones techniques implemented using waves2Foam, [4]. Absorption conditions were also implemented perimetrical of the computational domain to avoid the presence of reflected waves. Any solid boundary of the domain, named *front back*, *and bottom* and constructions called *geometria* inside the domain, are considered as plane, non-permeable, and described applying wall boundary conditions as provided by OpenFOAM®. Information about the implementation of wall-boundary conditions can be found in the work of Kalitzin et al. [26].

#### 3.2. InterFoam Settings

The implementation of the interfoam solver is performed adjusting the three basic files that are referred as controlDict, fvSchemes and fvSolution. The controlDict file contains simulation parameters such as the time step, which can be specified either as *fixed* or as *adjustable* to maintain Courant number, $C_0 = u_i \Delta t / \Delta x$, under a specific, defined by the user, value. In all simulations we use adaptive time stepping based on a maximum allowed CFL number of 0.15 as indicated by Larsen et al. [8]. Discretization schemes referring to the individual terms of the flow equations are given in the fvSchemes file. Time derivative $\partial / \partial t$ in the momentum equation e.g., is defined with the ddt scheme which can be selected among the available numerical schemes, *steadyState*, *Euler*, *Backwards* and *CrankNicolson*. A combined scheme, *Euler* with *CrankNicolson* was adopted in this work for the time derivative in the momentum equation, registered as *CrankNicolson* with a blending factor of 0.3, also proposed by Larsen et al. [8]. The fvSolution file contains the options of linear solvers and solution algorithms. Here, the user can define the iterative solvers, solution tolerance and algorithm settings. The available iterative solvers are PCG and PBiCG, preconditioned conjugate and (Bi-) gradient solver respectively, a smoothSolver, a generalized geometric-algebraic multi-grid solver denoted as GAMG and a diagonal solver. Specifically, the PCG solver is used to solve pressure equations *pcorr*, and *p_rgh*. The

pressure-velocity calculation procedure for the Navier-Stokes equations is achieved with the appropriate algorithm SIMPLE – PISO – PIMPLE, which can be specified also in the fvSolution file. In all simulations PISO loop is selected setting the entry nOuterCorrectors to value 1.

A common basic parameter for the three-dimensional physical applications is the significant computational cost. The spatial discretization is performed taking into account at least 10 grid points per wave height and maintaining the ratio 1:2 and 1:3 (depending on each simulation's characteristics) for the vertical to horizontal dimension of computational cells. Jacobsen et al. [4] suggest the use of a 1:1 ratio for two-dimensional applications. However, in this work the aspect ratio of numerical cells dimensions vertical to horizontal was kept at 1:2 and 1:3 values, rather than the 1:1, limiting the significant computational cost without loss in accuracy. However, the required small-time step (to keep Courant number $C_0 \leq 0.15$) in addition to the demand of at least 10 grid points per wave height require a great amount of computational sources. OpenFOAM-3.0.1 version is used for all the simulations.

## 4. Sediment Transport

The sediment transport in the coastal zone is divided into bed load, suspended load, and sheet flow. As the magnitude of the wave's orbital velocity on the seabed increases and as the waves approach the breaker line, these three processes are observed consecutively: bed-load movement, sediment suspension in the vicinity of bed ripples, and sheet-flow movement. In the surf zone, intense disorder of the water caused by wave breaking leads to higher sediment suspension. In the swash zone, sediment moves mainly as sheet flow under the action of uprush and downrush. The total time-averaged sediment transport rate (including pores), $\boldsymbol{q}_t = (q_{tx}, q_{ty})$, is estimated by

$$\boldsymbol{q}_t = \overline{\boldsymbol{q}_b} + \boldsymbol{q}_s \tag{12}$$

where $q_{tx}$, and $q_{ty}$ are the transport rates in directions $x$ and $y$, $\boldsymbol{q}_b = \left( q_{bx}, q_{by} \right)$ is the bed load transport and $\boldsymbol{q}_s = \left( q_{sx}, q_{sy} \right)$ is the time-averaged suspended load under broken waves.

Dibajnia and Watanabe [21] introduced a sheet-flow transport rate formula involving unsteady aspects of the sand transport phenomenon. Dibajnia [22] expanded the formula to take into account the bed load and the suspended load over ripples. In their works [23,27], the formula has been modified to estimate the sand transport rate under irregular sheet-flow conditions. According to the formula, in an asymmetric oscillatory flow, the total net sand transport rate (sheet flow, bedload, and suspended load over ripples) is essentially described as the difference between the two gross amounts of sand transported during the positive "crest" half-cycle and during the negative "trough" half-cycle. The sediment transport rates during the crest and trough half-cycle are related to the near bottom velocities (due waves and currents) which are calculated by the present wave model.

In a 2DH wave-current interaction field, total velocity field is demonstrated by the vector $\boldsymbol{u}_c$ which is equivalent to an interval of $\mathbf{T}_c$ followed by the vector $\boldsymbol{u}_t$ that is equivalent to an interval of $\mathbf{T}_t$. The sediment transport vector $\boldsymbol{q}_b$ is now estimated by [25,26].

$$\frac{\boldsymbol{q}_b}{w_s d_{50}} = a_{DW} \frac{\boldsymbol{u}_c T_c (\Omega_c + \Omega'_t) + \boldsymbol{u}_t T_t (\Omega_t + \Omega'_c)}{(T_c + T_t) \sqrt{(s-1)g d_{50}}} \tag{13}$$

where $w_s$ represents sediment fall velocity, $a_{DW}$ is the proportionality coefficient, $\boldsymbol{u}_c = (u_c, v_c)$ and $\boldsymbol{u}_t = (u_t, v_t)$ are the equivalent root-mean-square velocity amplitudes for the positive (crest) and negative (through) portions of the velocity profile.

The values of $\Omega_j$ are determined as in [16,22].

Experiments indicate that especially for highly asymmetric waves the sand that had been entrained during the positive cycle was brought back into the negative direction by the successive negative cycle. In some cases this mechanism (phase lag effect) is strong

enough to make the net transport be in the negative direction. The above Dibajnia and Watanabe formula takes into account the above mechanism.

According to Karambas and Koutitas [17], the time-averaged approach for estimating the suspended load induced by wave breaking can be assumed. Thus, the total submerged weight transport rate can be estimated by [17,28], using the equation:

$$q_s = \frac{1}{a} \frac{\varepsilon_s \overline{D} U_c}{w_s} \tag{14}$$

where $\overline{D}$ is the time average dissipation (the overbar denotes time averaging), $\varepsilon_s = 0.01$ is the suspended load transport efficiency factor, $U_c$ is the current velocitiy (estimated after time integration of the wave velocity as in [27] and $\alpha = (1-n)(s-1)\rho$., n is the porosity. The time average dissipation of the wave energy $\overline{D}$ is estimated by [29,30]:

$$\overline{D} = \overline{\rho h \varepsilon}, \quad \varepsilon = 0.09 k_b^{2/3}/l_e \tag{15}$$

where $\varepsilon$ is the dissipation of the near bottom wave turbulent kinetic energy, $k_b$ and $l_e$ is the scale of turbulence in the surf zone which is expressed as a function of the water depth, $h$; for example, $l_e = 0.2\ h$. The turbulent kinetic energy is calculated by the model results.

On the other hand, SEDFOAM, [30] also based on interFoam solver for the prediction of sediment transport rates is a distinguished attempt for a complete implementation of sediment transport predictions in the OpenFOAM® computational environment but has not yet been applied to real-scale coastal applications.

## 5. Validations

### 5.1. Wave-Induced Circulation around a Detached Breakwater

In coastal zones, wave propagation under the influence of shoaling and the presence of coastal structures, leads to the appearance of wave diffraction and wave breaking phenomena, resulting in water circulation. This sequence constitutes one of the main surf zones sediment transport mechanisms. In view of the importance of breakwaters for coastal engineering and their impact on coastal morphological changes, in this validation chapter, an experimental case is simulated using the interFoam solver. Numerical results are compared with experimental values concerning wave height and wave-induced currents measurements in the lee of a detached breakwater under the influence of harmonic waves. As far as the wave-induced current velocity field is concerned, two circulation currents were formed downstream of the breakwater. The wave-induced set-up/down at the exposed and the sheltered area drives the generation of local currents. These currents are directed towards the sheltered area from either side of the structure leading to the formation of two vortices. To ensure the ability of the numerical scheme to simulate this hydrodynamic mechanism, an experimental layout based on the Mory and Hamm [18] experiment was investigated concerning wave height and currents measurements in the lee of a detached breakwater under the influence of regular waves. The wave basin, 30 m by 30 m, consists of three parts. A 4.4 m long zone of constant depth h = 0.33 m close to the wave generator, an underwater 16.5 m long plane beach of 1/50 gradient, and an emerged plane beach of 1/20 gradient. A 6.66 m long and 0.87 m wide breakwater was built on the right side of the domain to produce diffraction phenomena under the influence of water waves. The numerical grid is intersected with the bathymetry surface using *snappyHexMesh* tool. Approximately, 27,932,616 cells were used to discretized 21.87 m at × (horizontal) direction, 11.97 m at *y* (horizontal), and the relative depth, considering 10 grid points per wave height. Simulation time is approximately 10 days using 8 cores (3.6 GHz) and a RAM of 24 GB for the 120 s of wave propagation. Mesh decomposition is handled by the scotch automatic method. The aspect ratio of cell sides, vertical to horizontal dimension, was 1:3, $\Delta x = \Delta z = 0.0225$ m, $\Delta y = 0.0075$ m. In the following Figure 1, the simulated area which is smaller at *z* direction compared to the experimental setup, the relative profile, and the position of relaxation zones are illustrated. In the simulations we adopted a shorter

dimension of the computational basin in *y* direction, in comparison with the *y* dimension of the experimental basin. This decreases the computational cost while at the same time does not affect the physical mechanism of the wave propagation and formulation. Model runs indicate that no wave reflection was observed from the wall the upper (*y* = 0) boundary of Figure 1, and, consequently, the position of the boundary does not contaminate the numerical solution. Water is simulated with density $\rho = 1000$ kg·m$^{-3}$ and kinematic viscosity, $\mu = 10^{-6}$ m$^2$·s$^{-1}$, while air has density $\rho = 1$ kg·m$^{-3}$ and the air kinematic viscosity $\mu = 1.48 \cdot 10^{-6}$ m$^2$·s$^{-1}$.

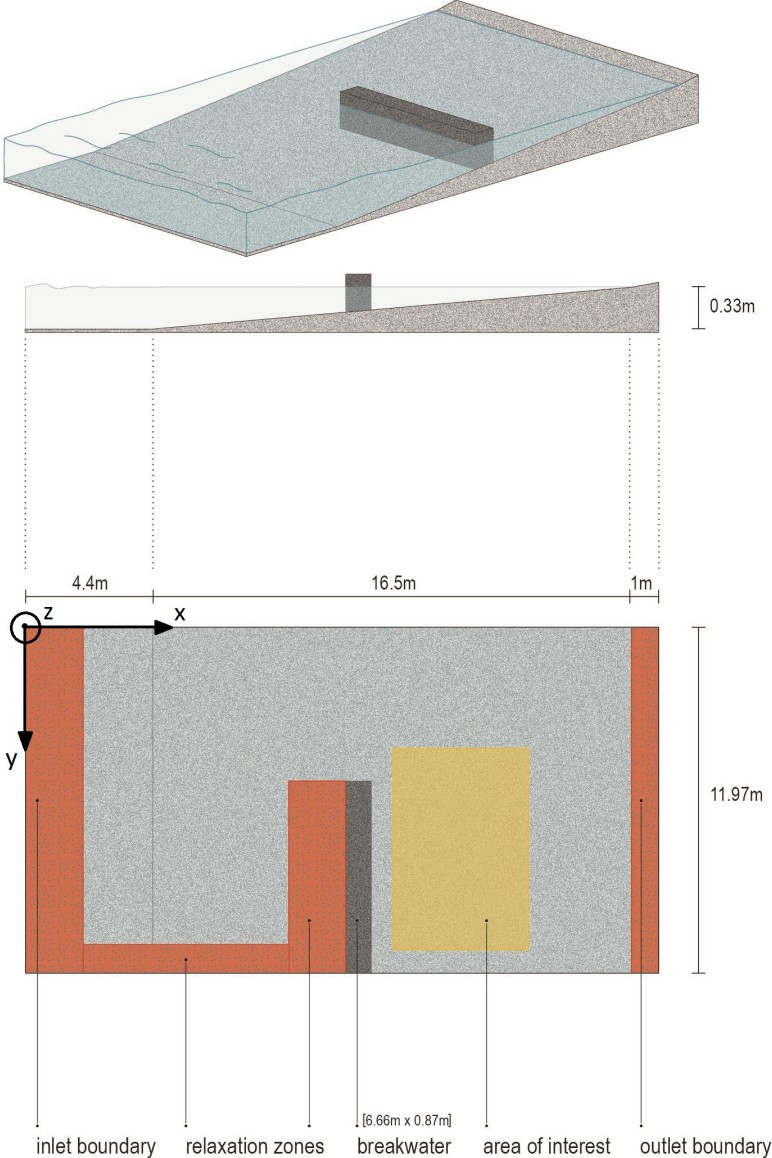

**Figure 1.** Numerical wave basin and relative geometric characteristics.

Regular waves of period *T* = 1.69 s and height *H* = 0.075 m, and are generated by Stokes first-wave theory at the inlet boundary. A relaxation zone is applied at the inlet boundary and just in front of the wavemaker of approximately one wavelength long to avoid reflected waves inside the domain. The bottom, the beach, the wave breaker sides, and the boundaries around the domain are considered as smooth solid walls and therefore no slip conditions have been applied for velocity, along with zero gradient (Neumann) conditions for pressure, and VOF fraction, *a*. Concurrently, wall functions are activated for *k* and $\omega$ at solid boundaries. Finally, for the atmospheric condition at the top of the computational domain, a combination of boundary conditions was used

for pressure, velocity, and volume fraction *a*, allowing water to flow out and air to flow into the domain when needed. Specifically, for pressure *p\**, *totalPressure* is used which is a *fixedValue* boundary condition calculated from the specified total pressure, $p_0$ and local velocity *U*. For velocity *U*, *pressureInletOutletVelocity* is used, which applies *zeroGradient* on all components, except where there is inflow, in which case a *fixedValue* condition is applied to the tangential velocity component and, finally, for volume fraction *a*, *inletOutlet* is used which is a *zeroGradient* condition when flow is outwards, *fixedValue* when flow is inwards. Wave reflection was not observed in the outlet because no water reached that boundary so a fixed wall boundary was also implemented at this boundary of the domain. In addition, relaxation zone is also applied and consequently, any fluid (water and air) reflection is absent. The model was run for seventy wave periods ~120 s, with the final thirty waves averaged and used for results. Using the tool waveGaugesNprobes, gauges were placed in the lee of breakwater (area of interest in Figure 1, in accordance with the locations of the wave gauges of the experiment) to measure wave height and current.

Model accuracy is investigated by wave height, along with current measurements in mean water depth, on the back side of the breakwater (denoted region in Figures 2 and 3). The results are compared against experimental values for regular wave conditions, *T* = 1.69 s and *H* = 0.075 m equally to experimental values. In Figure 2, the observed wave pattern is presented for regular waves after a sufficient number of wave periods propagation. Due to the presence of diffraction phenomenon, wave activity is reduced in that region. The breaking line was not clearly visible in the visualization picture (Figure 2), but this is only related with the visualization technique.

The wave height contour plot comparing numerical against experimental values is illustrated in Figure 3a. Wave height measurements were achieved using the *surfaceElevation tool* [4]. A good agreement is observed as numerical values are superimposed to experimental measurements for regular waves, with some overestimation located between *z* = 5–6 m and all along the distance from breakwater to shoreline. The wave height numerical values are satisfactorily consistent with experimental measurements, indicating the ability of the numerical model to simulate the three-dimensional formulation of wave propagation under the influence of breakwater and bottom presence. In Figure 3b, setup at two sections *x* = 21.0 m and *x* = 22.3 m (shoreline) is shown. Model results and experimental data are in good agreement.

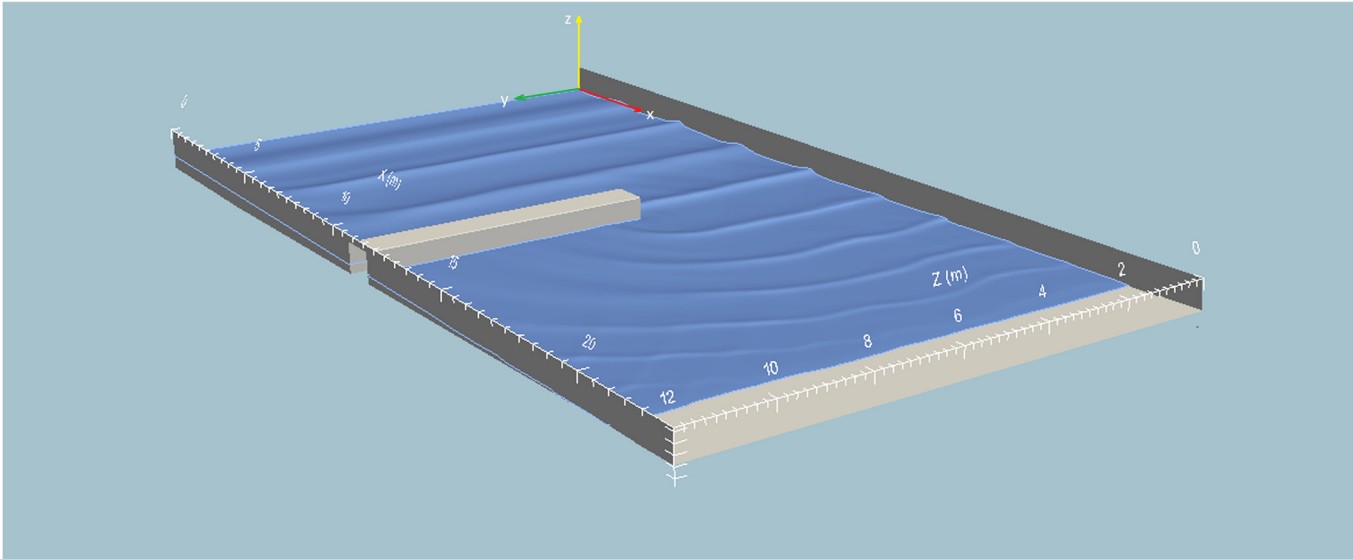

**Figure 2.** Free surface configuration of numerical simulation at the last time step, (*Paraview*® *visualization*).

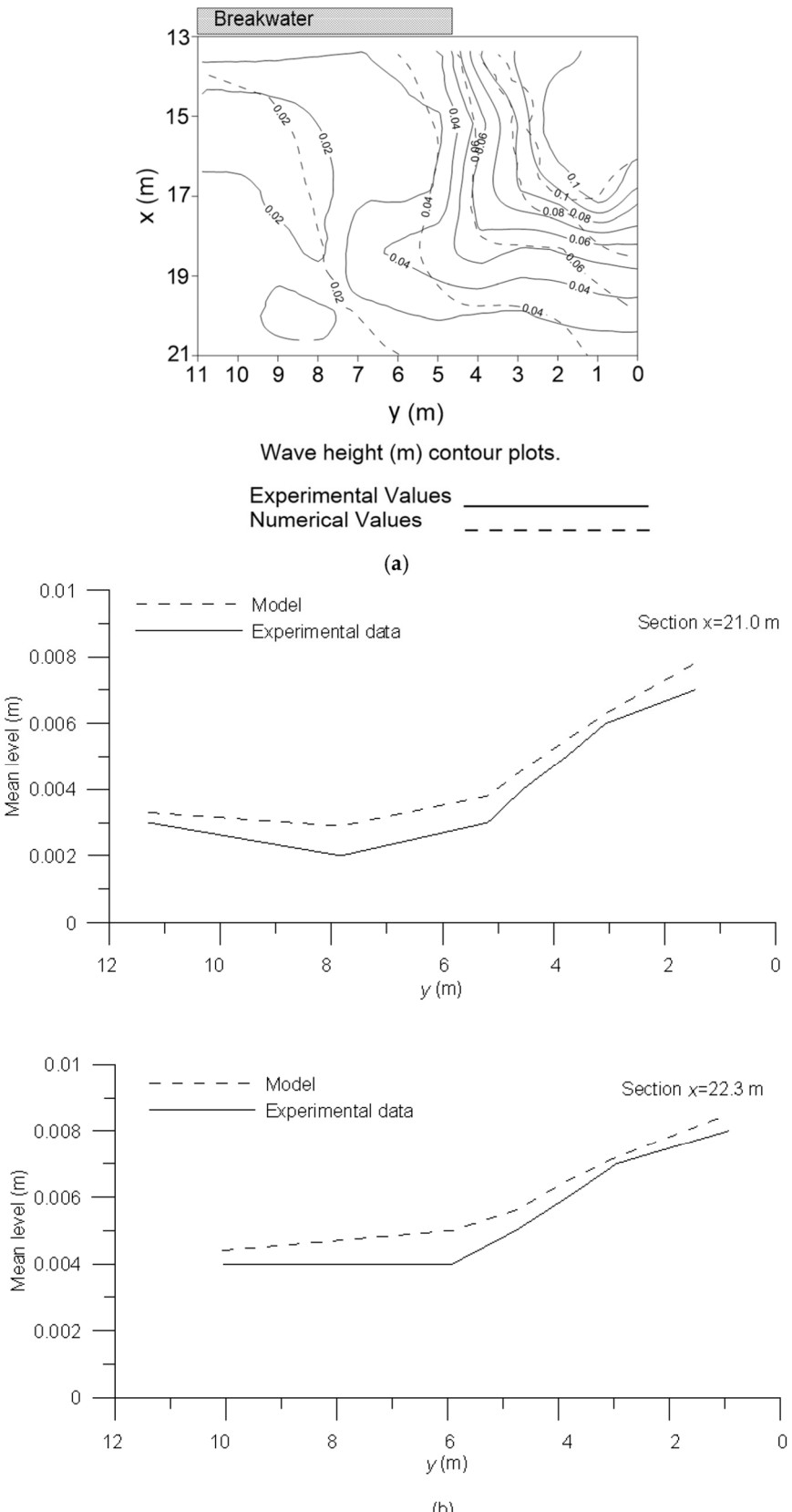

**Figure 3.** (**a**) Wave height distribution. Numerical solutions compared with experimental values. (**b**) Setup at sections *x* =21.0 m and *x* = 22.3 m (shoreline). Numerical solutions compared with experimental values.

The comparison between time-averaged velocities measurements at mid-water depth against experimental values is presented in Figure 4. Simulated time was extended enough (about 70 T~120 s) to achieve the stability of the hydrodynamic phenomenon. Eddy pattern was observed and measured behind the breakwater, producing a strong jet–like flow of up to 0.25 m/s, similar to the experimental results. A striking feature of numerical simulation is the wide eddy center with almost quiescent fluid which is located at 11.5 m < $y$ < 15.0 m and 27.0 m < $x$ < 28.5 m for the experiment and 5.5 m < $y$ < 9.5 m and 14.5 m < $x$ < 15.5 m for the numerical simulation. It is obvious that the quiescent fluid center is half of a meter displaced comparing with experimental layout, and has slightly smaller dimensions. As also indicated in Mory and Hamm [18], the position of quiescent center is the delimitation of the breaking line as eddy flow is driven by the wave breaking in surf zone.

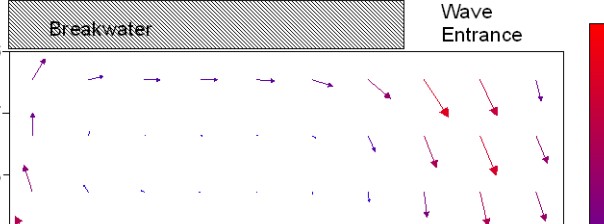

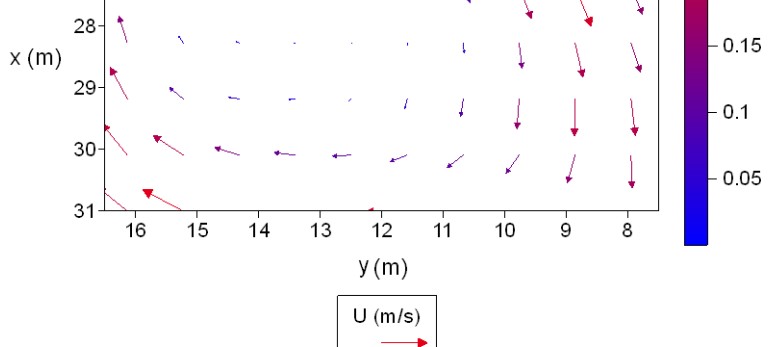

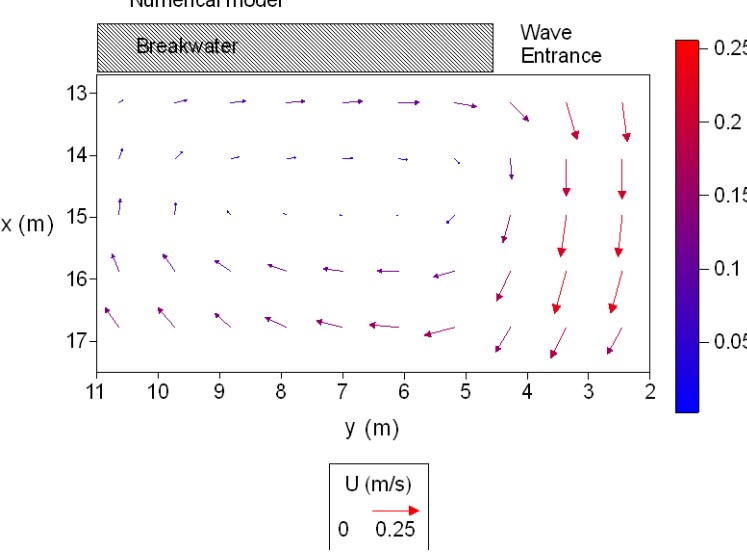

**Figure 4.** Time-averaged horizontal velocities at mid-water depth behind the breakwater: Comparison between numerical values (**below**), with experimental results (**above**) [18].

The numerical current pattern is satisfactorily consistent with the experimental layout, indicating that the numerical set up can reproduce significantly complex physical wave processes of coastal zones with respect to their three-dimensional characteristics. In addi-

tion, the numerical results are compared to the experimental ones by means of statistical indicators such as normalized root mean square error (NRMSE) defined as:

$$NRMSE = \sqrt{\frac{\sum_i^N (X_i - Y_i)^2}{\sum_i^N X_i^2}} \tag{16}$$

where $N$ is the sample size $X_i$ the experimental values and $Y_i$ the numerical results. In the above case (i.e., the current filed), the *NRMSE* is 28%.

### 5.2. Longshore Sediment Transport under Spilling and Plunging Breakers

For a complete evaluation and verification of longshore current and relative sediment transport rates, a part of the experimental work of Wang et al. [19] concerning large-scale laboratory measurements of longshore sediment transport under spilling and plunging breakers was simulated. Numerical results compared with experimental measurements in every case.

The experimental wave basin has dimensions of 30-m cross-shore, 50-m longshore, and maximum water depth of 0.9-m. The beach is composed of fine quartz sand with a median grain size of 0.15 mm. After a certain number of hours under the influence of incident wave conditions, the beach profile reached stable shape. This shape was created using *blockMesh* and *snappyHexMesh tool* (all included in OpenFOAM v3.0.1 suite) and was used as an input parameter for longshore current and longshore sediment transport rates predictions. The bathymetry is illustrated in Figure 5.

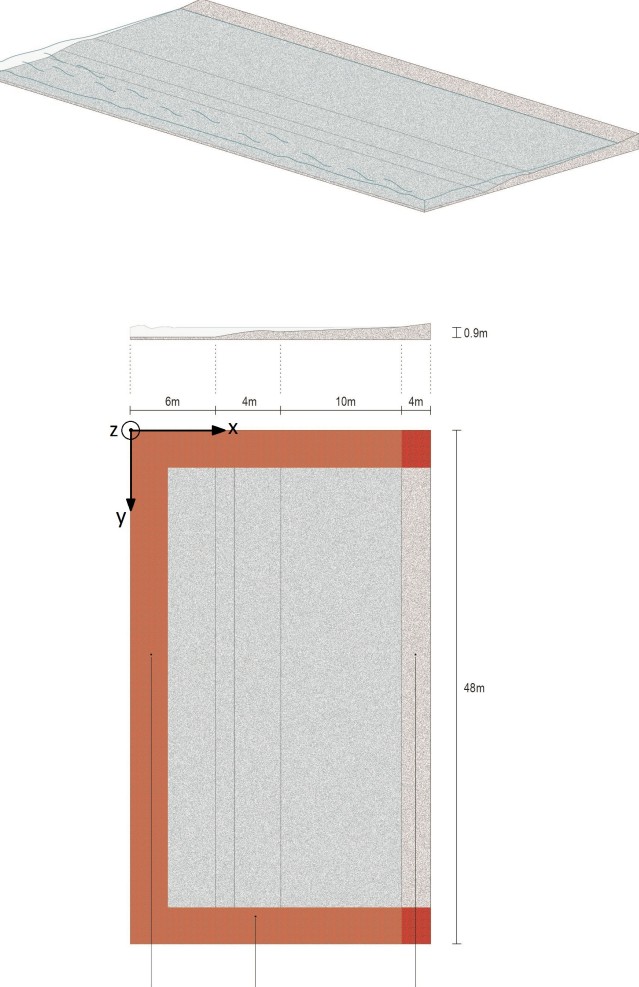

**Figure 5.** Numerical wave basin according to experimental layout and relative geometric characteristics.

For the experimental set-up, a TMA spectrum with spectral width parameter $\gamma$ equal to 3.3 was used to define the incident wave spectrum. For numerical simulations, a JONSWAP spectrum with the same characteristics was provided as an inlet boundary condition using wave2Foam. All other boundary conditions are used as in the previous validation case. Relaxation zones were used at the perimeter of the computational domain as seen in Figure 5, to prevent reflected waves from wall boundaries. Wave conditions for spilling and plunging breakers cases are given in Table 2.

**Table 2.** Simulated wave conditions.

| Plunging Breakers | |
|---|---|
| Significant wave height, $H_s$, (m) | 0.25 |
| Peak period, $T_p$, (s) | 3 |
| Wave angle (deg) | 10 |
| **Spilling breakers** | |
| Significant wave height, $H_s$, (m) | 0.23 |
| Peak period, $T_p$, (s) | 1.5 |
| Wave angle (deg) | 10 |

Considering 10 grid points to wave height, a total number of 9,830,400 cells were used to discretize the numerical wave basin which is illustrated in Figure 5, 48 m–longshore, 24 m croshore, and maximum water depth 0.9 m, (top to bottom of the numerical tank 1.2 m). Simulation time was approximately 6 days using 8 cores (3.6 GHz) and a RAM of 24 GB for 120 s of wave propagation (approximately 80 wave periods for spilling and 40 wave periods for plunging breakers case). Mesh decomposition is handled by the *scotch* automatic method. The aspect ratio of cell sides, vertical to horizontal dimension, was 1:3, $\Delta x = \Delta z = 0.075$ m, $\Delta y = 0.025$ m. Wave gauges were used in cross-shore alignment in three profiles, $y = 15$ m, $y = 25$ m, $y = 35$ m, to measure cross-shore distributions of free-surface elevation and longshore current as well as turbulent kinetic energy at an elevation of 1/3–1/2 of water depth from the bottom.

Because the suspended sediment is transported alongshore by the longshore current, the cross-shore distribution of longshore current has a significant influence on the patterns of longshore sediment transport. As it was mentioned before, three profiles at $y = 15$ m, $y = 25$ m and $y = 35$ m are investigated to ensure the constant value of the flow along shoreline. Only the values of centered profile ($y = 25$ m) are illustrated in the next figures. In Figure 6, a snapshot of the free-surface for both cases is demonstrated.

In Figures 7 and 8, the time-averaged cross-shore distribution of longshore currents (i) is presented along with the wave height (ii) measurements for the plunging and spilling breakers case, respectively. Numerical results are compared against experimental values [19] and are found to be in a relatively good agreement. Concerning the plunging breakers, the NRMSE is 22% for the longshore current and 9% for the wave height. For the spilling breakers, the NRMSE is 12% and 8%, respectively.



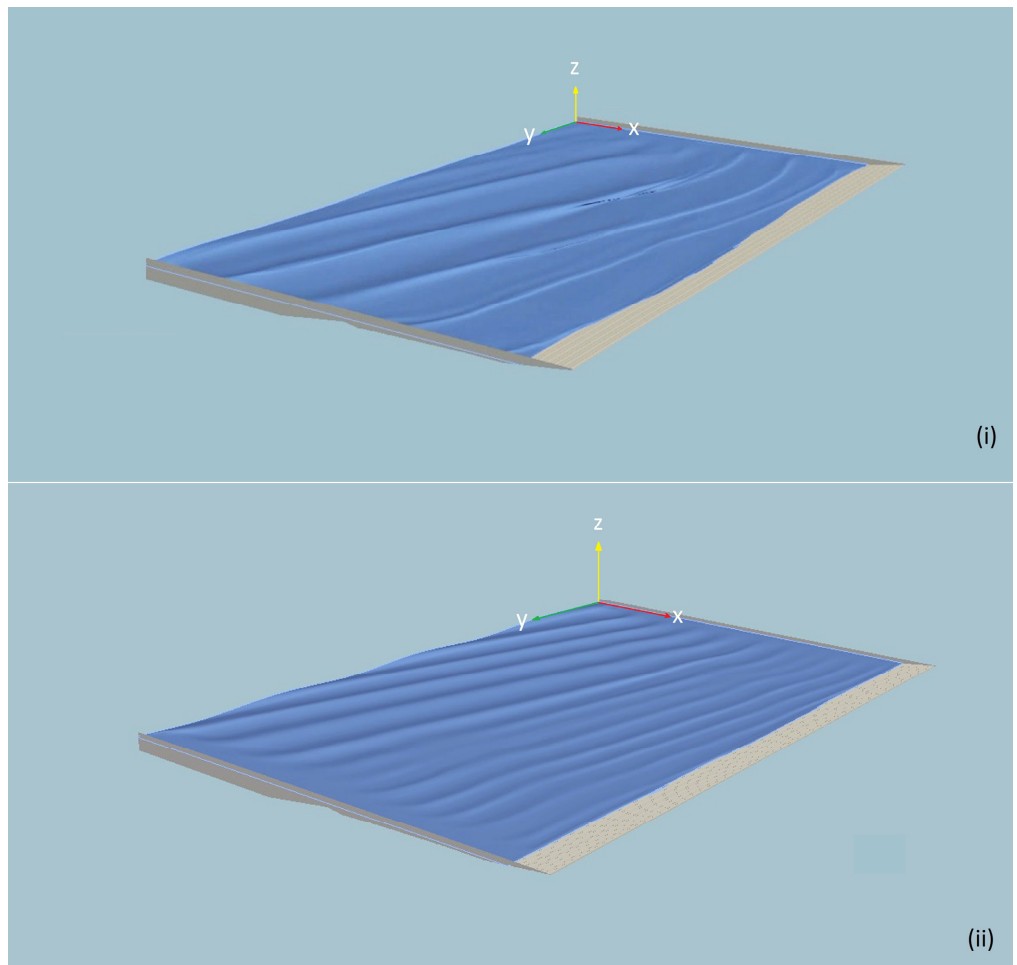

**Figure 6.** Free-surface configuration of numerical simulation at the last time step (**i**)-plunging breakers, (**ii**)-spilling breakers, (*Paraview*® *visualization*).

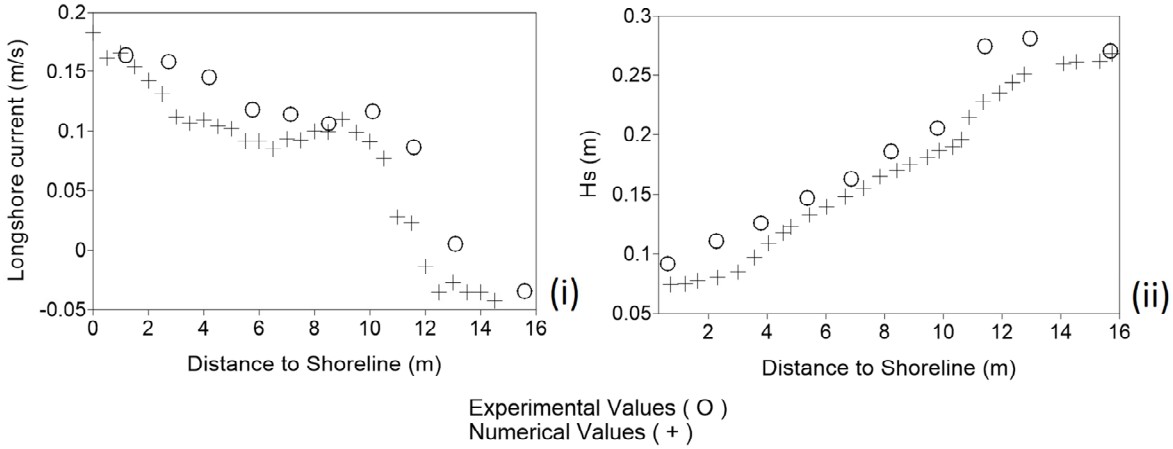

**Figure 7.** Time-averaged, cross-shore distribution (*y* = 25 m) of longshore current (**i**) and wave height (**ii**) measurements compared to experimental values [17] for the plunging breakers.

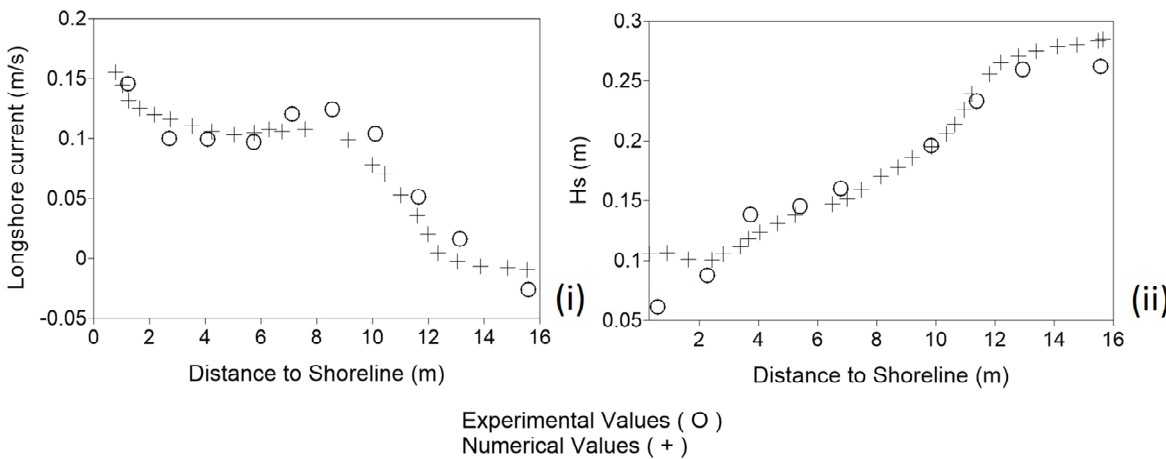

Experimental Values ( O )
Numerical Values ( + )

**Figure 8.** Time-averaged, cross-shore distribution ($y$ = 25 m) of longshore currents (**i**) and wave height (**ii**) measurements compared to experimental values [19] for the spilling breakers case.

In Figure 9, time-averaged turbulence kinetic energy (TKE) is presented for plunging (i) and spilling breakers (ii).

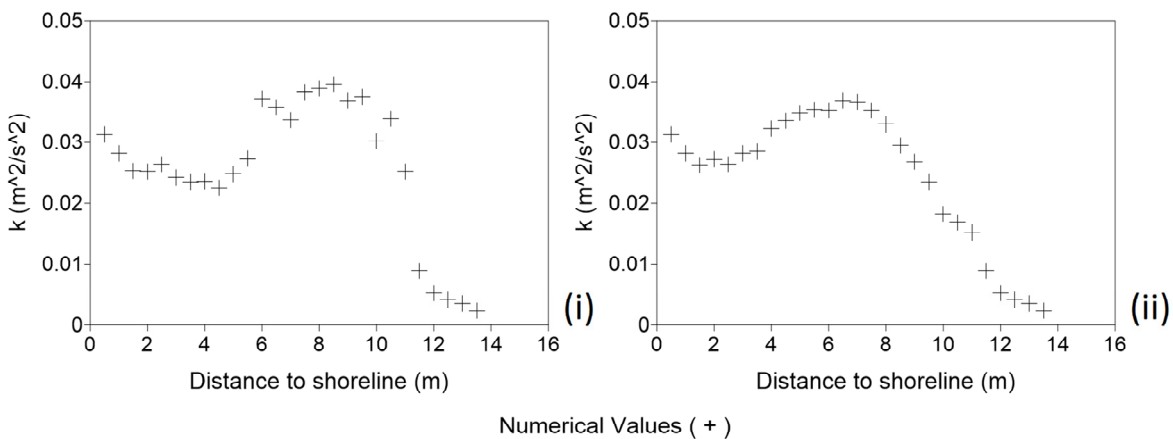

Numerical Values ( + )

**Figure 9.** Turbulence kinetic energy distribution for plunging (**i**) and spilling breakers case (**ii**).

According to Figure 7, wave breaking area for plunging breakers is located approximately 9 m from the shoreline, shortly after the position where the coastline material has been deposited, leading the beach profile to an equilibrium. The experimental measurements determine this point at 11.7 m as it is clearly shown in Figure 7ii, with an instant decrease in wave height. The numerical simulation estimates the breaking point slightly after 11 m from shoreline also indicating an underestimation of significant wave height values, about 10% reduced comparing to experimental measurements. Simultaneously, an instant increase is observed for TKE (Figure 9i) indicating the entrance of the surf zone for plunging breakers. Same observations are also taking place for the spilling breakers case, with more gentle variations either in wave height reduction or TKE increase (Figures 8ii and 9ii), respectively.

Generally, a good agreement is observed between numerical and experimental values concerning time-averaged cross-shore distribution of longshore currents in both plunging and spilling breakers (Figures 7i and 8i). Numerical results indicate that the generation of longshore currents is taking place in the breaker zone, just after 12 m from the shoreline (Figure 10). In the surf zone, the current has slightly constant values and it is getting its maximum value at the beginning of the swash zone, approximately 2–3 m from the shoreline. Overall, the small differences in magnitudes and patterns of longshore currents between the spilling and plunging breaker cases designate the independence of longshore

currents from different types of breakers. These results are also demonstrated in the experimental observation of [19].

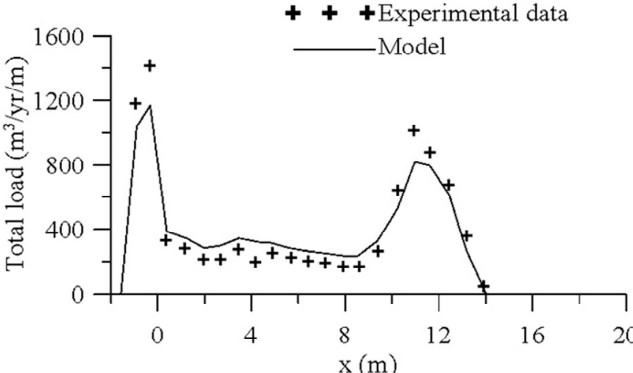

**Figure 10.** Comparison of sediment flux distributions for the plunging breaker case [19].

Taking into account all of the above hydrodynamic results, the numerical results concerning sediment transport rates are obtained and they are compared with the experimental data of [19] where longshore sediment transport rates under spilling and plunging breakers were studied. Figures 10 and 11 show comparisons between measured and predicted sediment flux distributions.

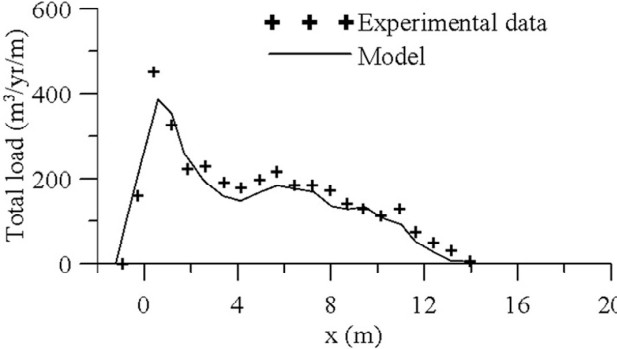

**Figure 11.** Comparison of sediment flux distributions for the spilling breaker case [19].

For both spilling and plunging breakers cases a non-uniform distribution of sediment flux cross-shore profile is observed. For spilling breakers, maximum sediment flux is observed inside swash zone as observed in Figure 11. For plunging breakers, two maximum regions of sediment flux are observed, either in swash or surf zone (Figure 10). In surf zone, the greater amount of turbulence kinetic energy due to wave breaking increases suspended sediment concentrations which are transported alongshore under the influence of longshore current. This fact indicates that the amount of suspend load is partly determined by breaker type. Swash zone's sediment flux values indicates the necessity of the implementation of the numerical scheme in this region as the highest values of cross-shore sediment transport rates are observed in this area, both for spilling and plunging breakers. The numerical results are in good agreement with experimental values. Some numerical underestimations for swash and surf zone maximum values of sediment flux are observed for both cases. The NRMSE is 17% and 16% for the plunging and spilling case, respectively.

## 6. Investigation of Longshore Sediment Transport Rates under Irregular Waves–Application in a Realistic Dimensions' Slopping Beach

### 6.1. Hydrodynamic Implementation

To ensure the applicability of the numerical scheme as a tool in realistic dimension coastal engineering application concerning the prediction of the hydrodynamic characteristic of the flow, a numerical wave basin of 148-m cross-shore, 340-m longshore dimensions,

and maximum water depth of 5-m was simulated under the influence of irregular waves. Exported results concerning cross-shore distributions of longshore current and turbulence kinetic energy were used to predict total longshore sediment transport rates. Kamphuis [31] empirical formula was used to validate the numerical results.

Considering 10 grid points to wave height, a total number of 10,064,000 cells were used to discretize the numerical wave basin which is illustrated in Figure 12: 340 m–longshore, 148 m cross-short, and maximum water depth 5 m (top to bottom of the numerical tank 6.4 m). Simulation time is approximately 7 days using 8 cores (3.6 GHz) and a RAM of 24 GB for 160 s of wave propagation, approximately 20 $T$. Mesh decomposition is handled by the *scotch* automatic method. The aspect ratio of cell sides, vertical to horizontal dimension, was 1:2, $\Delta x = \Delta z = 0.4$ m, $\Delta y = 0.2$ m. An adjustable time step was used to keep Courant number, $C_0 = u_i \Delta t / \Delta x_i \leq 0.15$. Wave gauges were used in cross-shore alignment in $y = 200$ to measure the free-surface elevation and the cross-shore distribution of longshore currents as well as the turbulent kinetic energy, at an elevation of 1/3–1/2 of water depth from the bottom. A JONSWAP spectrum with spectral width parameter $\gamma$ equal to 3.3 was provided as an inlet boundary condition using wave2Foam. All other boundary conditions were used as in the previous validation case. Relaxation zones were sited at the perimeter of the computational domain to prevent reflected waves from wall boundaries. Wave conditions are presented in Table 3.

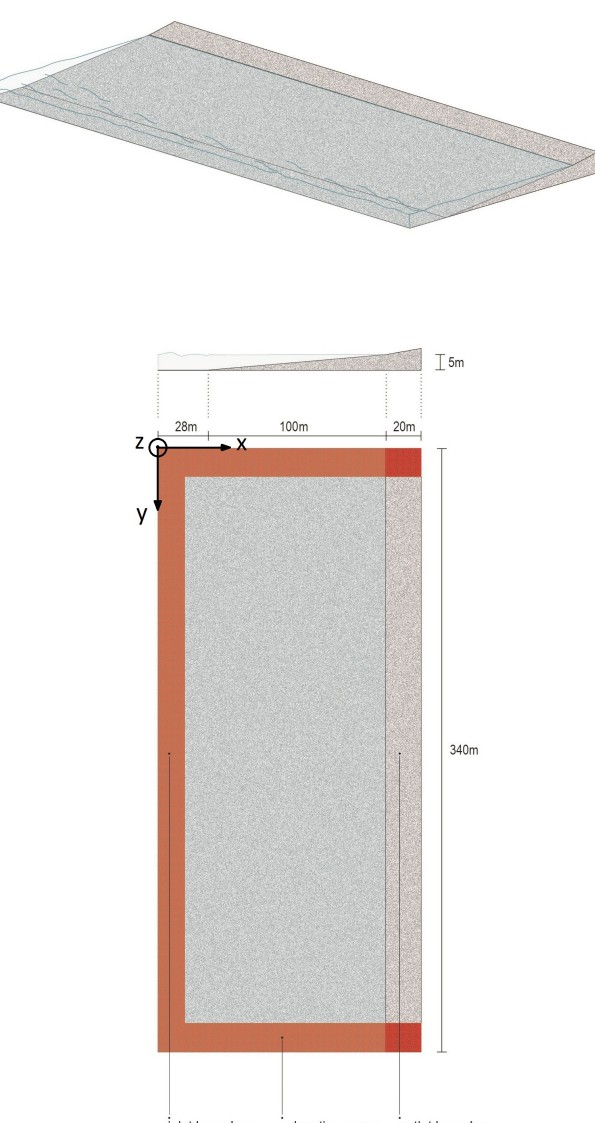

**Figure 12.** Numerical wave basin and relative geometric characteristics.

**Table 3.** Wave conditions.

|  | **Wave Condition 1** | **Wave Condition 2** | **Wave Condition 3** |
|---|---|---|---|
| Significant wave height $H_s$, (m) | 2 | 3 | 2 |
| Peak period, $T_p$, (s) | 8 | 8 | 8 |
| Wave angle $\theta$ (deg) | 26 | 32 | 38 |

*6.2. Results*

Last 10 wave periods were time-averaged to obtain results concerning cross-shore distribution of longshore currents and turbulence kinetic energy TKE at $y = 200$ m. In Figure 13, surface elevation patterns for each wave condition are illustrated. In Figure 14, the results are presented for each wave condition according to Table 2.

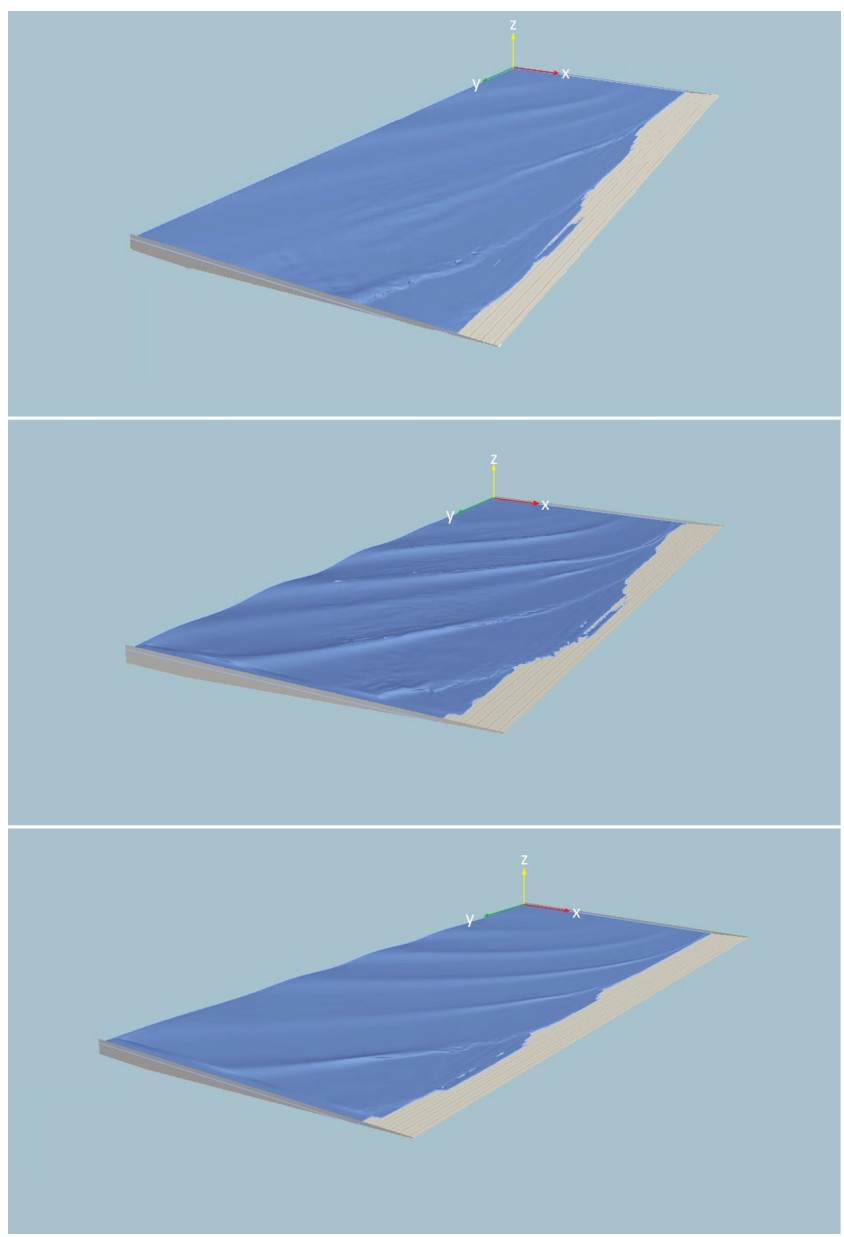

**Figure 13.** Free-surface configuration of numerical simulation at the last time step for each wave condition. Top-wave condition 1, center-wave condition 2, bottom-wave condition 3-(*Paraview*® *visualization*).

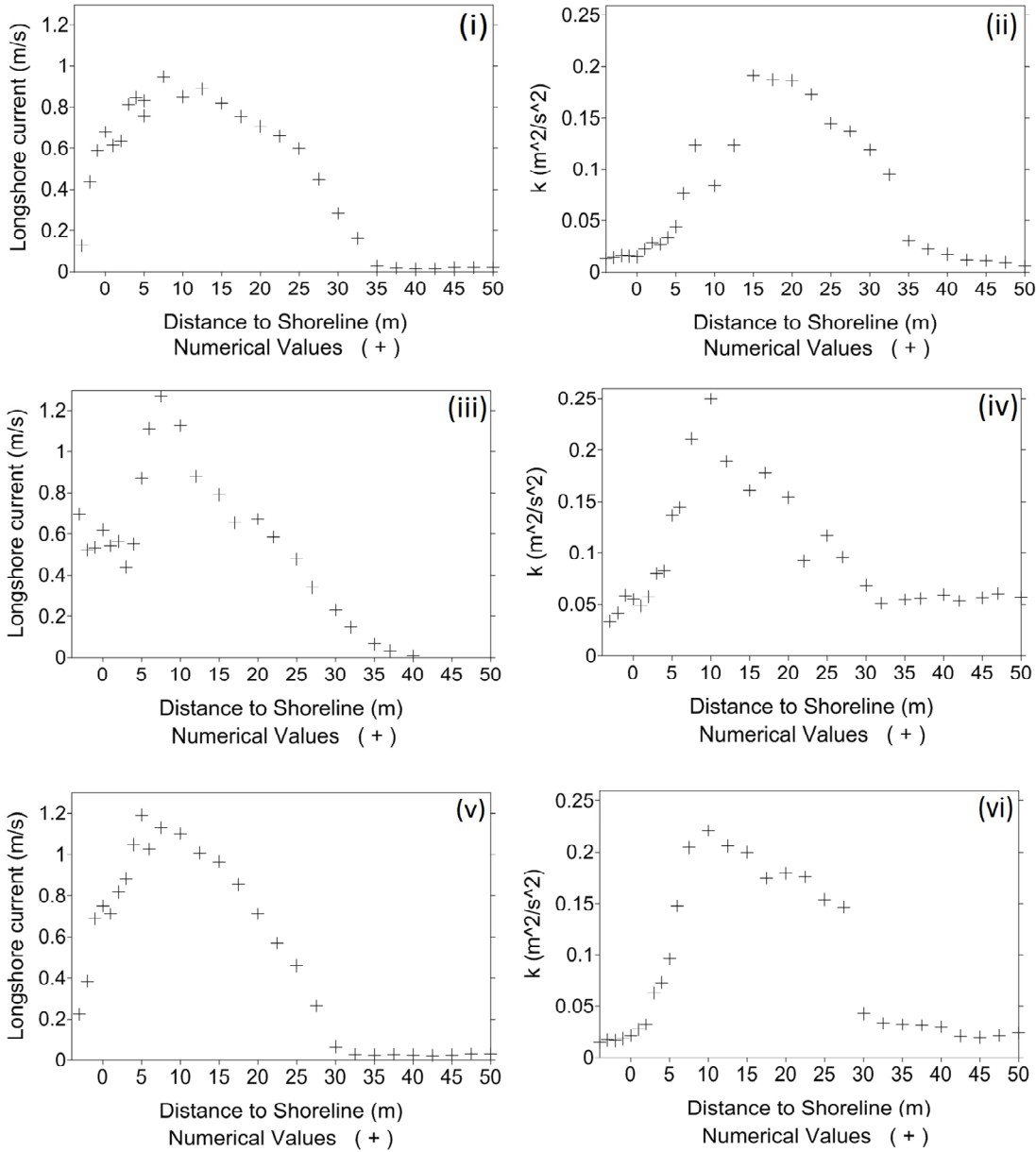

**Figure 14.** (**i–vi**). Cross-shore distribution of longshore current and turbulence kinetic energy for each wave condition. Figures (**i,ii**)-Wave condition 1, Figures (**iii,iv**)-Wave condition 2, Figures (**v,vi**)-Wave condition 3.

For a complete evaluation of hydrodynamic and sediment transport rate processes, the exported results are also obtained in the landward side of the swash zone approximately −5 m from the shoreline. Current measurements indicate a maximum value in the seaward side of the swash zone, approximately 5–10 m from shoreline, for all wave conditions (Figure 14i,iii,v). It is obvious that the increase of wave angle in the wave generator leads to greater current values. In the first wave condition ($\theta = 26°$), averaged velocities up to 1 m/s are observed, indicating the existence of a longshore current generated in the entrance of the surf zone, approximately 25 m from the shoreline (Figure 14i). Simultaneously, turbulence kinetic energy increases rapidly up to 0.2 m²/s² inside the surf zone, indicating the existence on turbulence due to wave breaking. A similar pattern is observed for the other two wave conditions with steeper variation of current, wave height and TKE measurements as wave-group propagates. Specifically, as the wave angle in wave generator takes its maximum value of 38°, the currents' measurement indicates values up to 1.2 m/s (Figure 14v). For a smaller wave angle of 32° (wave condition 2) but for increased significant

wave height to 3 m (from 2 m), current values reach their maximum value of 1.25 m/s (Figure 14iii), indicating the greater influence of wave height rather than the wave direction in the presence of longshore currents. It is worth mentioning that the current measurements indicate the existence of transport rates all inside the swash zone, even in the landward side, up to 0.6 m/s.

### 6.3. Longshore Sediment Transport in a Plane Beach

In this paragraph, numerical results concerning longshore sediment transport rates obtained using the above formulation. Moreover, the exported numerical values are compared against Kamphuis [31] empirical formula for longshore sediment transport estimation. The volumetric total longshore transport rate $Q_y$, from the swash zone across the surf zone to deep water, is calculated by the cross-shore integration of the longshore transport rate $q_{ty}$ ($q_{ty} = q_{by} + q_{sy}$). Model results are compared with the Kamphuis [31] empirical formula which is based on experimental and field data for the estimation of total alongshore transport rate. The formula is written:

$$Q_t = 0.0023 H_{sb}^2 T_P^{1.5} (\tan \beta)^{0.75} d_{50}^{-0.25} \sin^{0.6}(2a_b) \left( \mathrm{m}^3/\mathrm{s} \right) \tag{17}$$

where $H_{sb}$ is significant breaker height, $T_p$ is the peak period, *tanβ* is the surf zone slope, and $a_b$ is the breaker angle. In the Equation (17), the significant breaker height $H_{sb}$ and the breaker angle $a_b$ are estimated from the present model. In the first numerical experiment (wave condition 1, Table 2) the following computational conditions are assumed: peak wave period of the incident spectrum $T_p = 8$ s, uniform slope $\tan \beta = 1/20$, and incident angle $a_b = 25°$ at the breaking point. In Figure 15, the predicted longshore transport rates (in $\mathrm{m}^3$/s) for different grain sizes are plotted against deep water root-mean-square wave height $H_{0\_rms}$. In comparison with the well-confirmed Kamphuis [31] formula, the present calculations generally over predict the total longshore transport, especially for high waves. Model results seem to be more sensitive to variation in the incident wave height. It is mentioned here that the empirical nature of the formula (including uncertainties in all involved parameters) does not permit for a strict qualitative comparison between model results and the formula predictions.

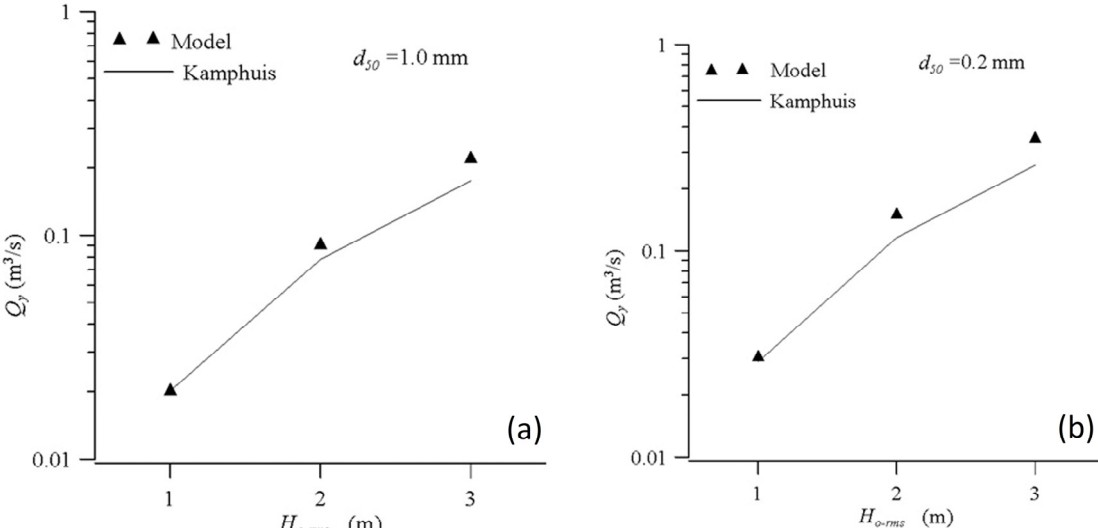

**Figure 15.** Comparison between calculated longshore transport rate and Kamphuis formula [31] for different deep water incident wave heights $H_{o\text{-}rms}$: sediment median diameter (**a**) $d_{50} = 1.0$ mm and (**b**) $d_{50} = 0.2$ mm.

In Figure 16, comparisons between calculated longshore transport rates are also shown, for different peak wave periods $T_p$ and two sediment median diameters $d_{50}$ ($d_{50} = 0.2$ mm and $d_{50} = 1.0$ mm). The following computational conditions are also assumed: deep water

wave height $H_{o\text{-}rms}$ = 1.0 m, uniform slope tan $\beta$ = 1/20, and incident angle $a_b$= 25° at the breaking point.

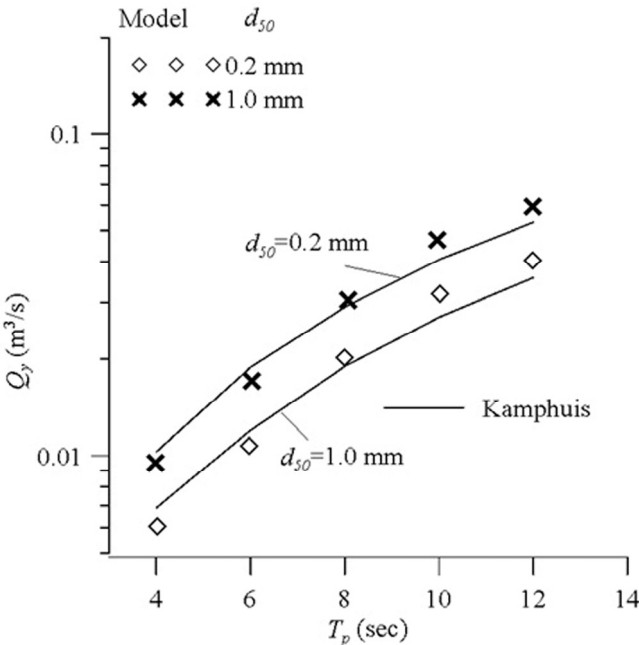

**Figure 16.** Comparison between calculated longshore transport rate and Kamphuis [31] formula for different peak wave periods $T_p$ and sediment median diameters: $d_{50}$ = 0.2 mm, $d_{50}$ = 1.0 mm.

The Kamphuis formula [31] predicts total sediment rates well for different breaker types (spilling and plunging) as compared to the measured values by incorporating the wave period to a power of 1.5. The wave period also seems to have considerable influence on the range of up-rush and down-rush, which in turn influences the transport rate in the swash zone. The present model seems to be more sensitive to variation in the peak wave period. Finally, another comparison is presented in Figure 17: model predictions against Kamphuis formula [31] for different breaking angles. Again, the model predictions and the Kamphuis formula [31] agree quite well.

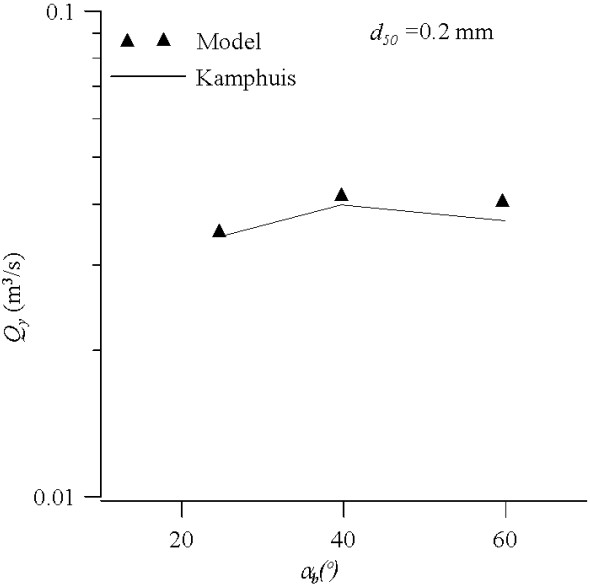

**Figure 17.** Model predictions against Kamphuis formula [31] for different breaking angles.



## 7. Conclusions

In this work, the applicability of the numerical model consisting of the interFoam multiphase solver, part of OpenFOAM®, coupled with waves2Foam and $k$–$\omega$ SST buoyancy modified turbulence model was investigated as a tool for the prediction of coastal zone hydrodynamics. Results concerning wave height, wave-induced currents, and Turbulence Kinetic Energy were compared with experimental values by the simulation of benchmark experimental cases of coastal engineering research [18,19]. All simulations ware executed in full three-dimensional (3D) applications with respect to the physical characteristics of wave processes such as wave breaking and wave-induced currents. In addition, exported hydrodynamic measurements were used to predict longshore sediment transport rates using the Dibajnia and Watanabe [19–21,25] mathematical formula. Numerical results indicate the ability of the combined numerical scheme to predict coastal zone hydrodynamics with accuracy. The prediction of longshore sediment transport rates was estimated using the well-known numerical formula of Dibajnia and Watanabe [21–23,27] with accurate results. The numerical setup was achieved by considering extremely useful previous works such as [8,9,14].

The combined numerical scheme seems to be suitable for the prediction of the hydrodynamic process of a coastal zone. The accuracy in wave characteristics and current measurements in real scale applications indicates the potential of the numerical suite for a wide-range implementation of realistic problems of coastal engineering. Especially in problems concerning prediction of sediment transport rates, the well-known numerical formulas seem to predict transport rates with accuracy.

The aspect ratio of numerical cells dimensions vertical to horizontal was kept at 1:2 and 1:3 values, which is used in the present work, to reduce the computational cost without loss in accuracy. However, the required small-time step (to keep courant number $C\_0 \leq 0.15$) in addition to the demand of at least 10 grid points per wave height require a great number of computational sources.

It was not part of this work to bring out the effect of individual solver parameters to the accuracy of results concerning the numerical methods or the algorithms for the manipulation of transport equations etc., as these attempts have been implemented in previous state-of-the-art works mentioned above. This work focuses on the application of a specific numerical combination for the prediction of coastal zone hydrodynamic and morphodynamic processes, proposing it as a solution for the manipulation of such kind of works.

**Author Contributions:** Conceptualization, I.K. and T.V.K.; methodology, I.K. and T.V.K.; validation, I.K.; writing—original draft preparation, I.K.; writing—review and editing, I.K. and T.V.K.; visualization, I.K.; supervision, T.V.K. All authors have read and agreed to the published version of the manuscript.

**Funding:** This research received no external funding.

**Institutional Review Board Statement:** Not applicable.

**Informed Consent Statement:** Not applicable.

**Data Availability Statement:** Not applicable.

**Conflicts of Interest:** The authors declare no conflict of interest.

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
