# Peer review of "Numerical Simulation of Hydrodynamics and Sediment Transport in the Surf and Swash Zone Using OpenFOAM®"

_jmse, doi:10.3390/jmse11020446_

Round 1

Reviewer 1 Report

Authors aim to demonstrate the effectiveness of the interFoam (embedded in the OpenFoam C++ toolbox) two-phase flow solver coupled with the modified k-ω SST turbulence closure model and with the wave2Foam waves generator, as a useful tool for the prediction of coastal zone hydrodynamics (in real scale applications).  The work is of numerical nature and the obtained results were validated against those obtained, experimentally, by other authors [(Mory et al. (1997) and Wang et al. (2002)]. Furthermore, to predict the sediment transport rate due to waves and currents, a transport rate formula is implemented.

The work is certainly interesting, but there are some shortcomings that must be addressed by the authors before continuing with the review process.

In the Introduction section the cited literature is quite relevant to the study. The problem to be investigated is clearly stated. As regarding the numerical model description section, I am not sure if the provided information are sufficient so that the simulations could be reproduced independently. Considering the fact that authors performed complex numerical simulations, the section is too weak and maybe does not reflect the work behind it. The text between lines 126 and 143, describing the well-known VOF algorithm, should be deleted. In my opinion other detail, for the specific case should be highlighted in the text. The turbulence model adopted for the simulations is the k-ω SST (right choice), but the considered equations (2.9 and 2.10) are not well described in the main text. What about functions F ? What about the term G in the equation 2.10 ? Are you referring to buoyancy term Gb also here ? What about values of the other coefficients ?   

Numerical setup description needs to refer a detailed sketch of the computational domain that should contain all the dimensions reported in the main text.

With regard to the bed load transport term (equation 2.12 and line 237), is this a time averaged term?

The sediment transport model section represents the core of the work and need a more detailed description; in particular, the implementation of the model into the numerical code (some words about SEDFOAM are reported only in the conclusions section) has to be clarified.

Another comment is related to the effect of the surface tension, did you analyze its effect in your simulations?

Between lines 320-322, Authors stated “Wave reflection was not observed in the outlet because no water reached that boundary, so a fixed wall boundary was also implemented at this boundary of the domain.”, I am not sure if this is a right choice by considering the fact that Authors are in the case of a multiphase flow (air and water): Authors are kindly invited to clarify. What about the influence of boundary conditions on the analyzed phenomenon ? At what distance were the numerical wave gauges probes placed ?

Another concern is related to the grid sensitivity analysis, any grid-refinement study carried out ? Some words on this are needed, at least.

Minors

Lines 90-98. This part should be rewritten considering the sections actually reported in the main text.

All the equations reported in the main text have two different numbers (ex. (2.1) and (1)), these must be standardized.

Between lines 102 and 103 Authors stated “The flow equation consists of the Continuity Equation and the Navier Stokes Equations…”, rephrasing considering that the Navier-Stokes equations consist of a time-dependent continuity equation for conservation of mass, three time-dependent conservation of momentum equations and a time-dependent conservation of energy equation.

Line 152. “Rusche, 2002” missing in the references section.

Line 156. “Higuera et al. (2013)”, (a) and/or (b)?

Line 187. Replace P_rgh with p_rgh.

Line 241. Check reference “Watanabe and Dibajnia [20]”.

Figure 1 is not clear, add reference axis.

Line 314. Is p* the pseudo-dynamic pressure defined at line 112?

Figure 4, check the year of publication of Mory and Hamm.

Figure 5 is not clear, add reference axis.

Figure 12 is not clear, add reference axis.

Line 630 and 661. Check the reference “Jacobsen et al. (2012)”. Is this the same reference?

Author Response

The authors would like to thank all reviewers for their constructive comments, which have largely improved the quality and readability of present manuscript. We have carefully considered all the remarks and make changes to the paper stemming from these comments, which are detailed below. 

For the reply to your comments please see attached file.

Author Response

(The authors gave the same response as above.)

Reviewer 3 Report

The authors presented a numerical investigation on the 3D hydrodynamic processes of coastal zones such as wave breaking, wave-induced currents and sediment transport, using OpenFOAM. The simulated results are compared against experimental data. The work adds an example to simulate the nearshore processes through the CFD method. I recommend the manuscript be accepted after a major revision. The specific points are as follows.

(1) The actual structure of the manuscript is grossly inconsistent with what is mentioned in the introduction. The validation, results and conclusions are respectively sections 3, 4 and 5 in introduction, while they are actually sections 5, 6 and 7 in the manuscript. It is recommended that sections 2, 3 and 4 be merged into a methodology as mentioned in the introduction.

(2) Line 50-89: Mainly an overview of the turbulence model is introduced in the introduction, the authors should also add the literature review related to the 3D processes of coastal zones such as wave breaking, wave-induced currents and sediment transport.

(3) Line 50-89: Larsen and Fuhrman (2018) indicated that the buoyancy production term included by Devolder et al. (2017; the buoyancy modified k–ω SST model) did not solve the fundamental problem of high turbulence. So, Larsen and Fuhrman (2018) developed a new formulation of the k–ω closure. Then, Li, Larsen and Fuhrman (2022) found out the unprecedented accuracy of the stress–ω model. The author should explain why they choose the buoyancy modified k–ω SST model and not the new stress–ω model.

(4) Line 120: The units should be unified by using “m-1” or “/m” in Table 1.

(5) Line 290: “27.932.616 cells” should be changed to “27,932,616 cells”

(6) Line 268-370: The Mory and Hamm experiment presented the horizontal variation of wave height, set-up and vertical profile of velocity. I think the authors should not only verify the wave height distribution but also set-up and velocity profile.

(7) Line 538-580: In section 6.3, the authors compared the simulated longshore sediment transport rates with the empirical formula of Kamphuis. More empirical formulas such as van Rijn (2014) should be compared?

(8) Line 608-615: The simulation of the 3D processes is a highlight of this paper, but the specific aspect ratio and courant number which reduce the computational cost are best described in detail in the main text rather than in the conclusion. Please simplify the presentation of this paragraph.

Author Response

(The authors gave the same response as above.)

Round 2

Reviewer 1 Report

The authors have satisfactorily responded to all my concerns and made the necessary changes to the manuscript. However, there are still some minor issues to be addressed.

Between lines 106-107, authors stated: “coupled with Volume of Fluid (VOF) Method”, this point is redundant as they discussed about VoF also between lines 132-133. I suggest to improve this part a little more! (Could be useful for authors analyze the following paper: Lauria and Alfonsi (2020). Numerical Investigation of Ski Jump Hydraulics. J. of Hydr. Eng. ASCE).

Between lines 253-256 authors stated: “The formula is applied to individual waves of a near bottom velocity time series which are calculated by the present model. The total net transport rates are calculated by taking the difference between the rates during successive half wave cycles.” This sentences should be more clear.

Between lines 320-322, authors stated: “The shorter y direction, we adopt here, decreases the computational cost while at the same time does not affect the physical mechanism of the wave propagation and formulation  since no wave reflection was observed from the y=0 wall boundary”. For me is not clear enough, please rephrase.

Check citations numbering in the main text.
